# Physical Activity and Quality of Life in Hemodialysis Patients and Healthy Controls: A Cross-Sectional Study

**DOI:** 10.3390/ijerph18041978

**Published:** 2021-02-18

**Authors:** Tjaša Filipčič, Špela Bogataj, Jernej Pajek, Maja Pajek

**Affiliations:** 1Faculty of Education, University of Ljubljana, Krdeljeva ploščad 16, 1000 Ljubljana, Slovenia; tjasa.filipcic@pef.uni-lj.si; 2Department of Nephrology, University Medical Centre Ljubljana, Zaloška cesta 2, 1000 Ljubljana, Slovenia; spela.bogataj@kclj.si (Š.B.); jernej.pajek@mf.uni-lj.si (J.P.); 3Faculty of Sport, University of Ljubljana, Gortanova ulica 22, 1000 Ljubljana, Slovenia; 4Faculty of Medicine, University of Ljubljana, Vrazov trg 2, 1000 Ljubljana, Slovenia

**Keywords:** quality of life, hemodialysis patients, physical functioning, impact, human activity profile

## Abstract

Hemodialysis (HD) patients have lower functional abilities compared to healthy people, and this is associated with lower physical activity in everyday life. This may affect their quality of life, but research on this topic is limited. Therefore, the present study aimed to determine the relationship between habitual physical activity and quality of life in HD patients and healthy controls. Ninety-three HD patients and 140 controls participated in the study. Quality of life was assessed using a 36-item medical outcomes study short-form health survey (SF-36). Human Activity Profile (HAP) was used to assess habitual physical activity. The adjusted activity score (AAS) from HAP, age, gender, fat tissue index (FTI), lean tissue index (LTI), and Davies comorbidity score were analyzed as possible predictors of the Physical Component Summary (PCS) of the SF-36. Three sequential linear models were used to model PCS. In Model 1, PCS was regressed by gender and age; in Model 2 the LTI, FTI, and Davies comorbidity scores were added. Model 3 also included AAS. After controlling for age and gender (Model_HD_ 1: *p* = 0.056), LTI, FTI, and Davies comorbidity score effects (Model_HD_ 2: *p* = 0.181), the AAS accounted for 32% of the variation in PCS of HD patients (Model_HD_ 3: *p* < 0.001). Consequently, the PCS of HD patients would increase by 0.431 points if the AAS increased by one point. However, in healthy controls, AAS had a lower impact than in the HD sample (B = 0.359 vs. 0.431), while the corresponding effects of age and gender (Model_H_ 1: *p* < 0.001), LTI, FTI, and Davies comorbidity score (Model_H_ 2: *p* < 0.001) were adjusted for. The proportion of variation in PCS attributed to AAS was 14.9% (Model_H_ 3: *p* < 0.001). The current study results showed that physical activity in everyday life as measured by the HAP questionnaire is associated to a higher degree with the quality of life of HD patients than in healthy subjects. Routine physical activity programs are therefore highly justified, and the nephrology community should play a leading role in this effort.

## 1. Introduction

Patients with end-stage renal disease (ESRD) treated with hemodialysis (HD) report lower physical functioning and lower quality of life compared to the healthy population [1,2,3]. Impaired physical functioning affects overall health and is associated with a reduced survival rate in this population [4,5,6].

The scientific literature shows that physical inactivity is a common phenomenon in HD patients, although exercise has been shown to be safe and to have numerous positive effects in those patients [7,8,9,10]. Given that HD patients report comorbidities associated with physical inactivity, determining the physical activity level is crucial for planning interventions to increase their mobility [1]. A variety of questionnaires are available to assess physical activity. The Human Activity Profile questionnaire (HAP) showed a strong correlation with other physical activity questionnaires/scores, including the International Physical Activity Long Questionnaire (IPAQ), the Four Week Physical Activity History Questionnaire (FWH), and the Physical Activity Scale for the Elderly (PASE) [11]. HAP has been widely used in the HD population, as it covers a wide range of activities compared to other questionnaires and assesses even the activities from the lowest spectrum [12].

It has been shown that physical functioning is associated with health-related quality of life in HD patients [13,14,15,16]. Reduced physical functioning and lower muscle strength affect their ability to perform activities of daily living and therefore impact their quality of life [17,18]. A recent study [19] in HD patients showed that attaining a satisfactory level of physical activity can contribute to a greater perception of quality of life. In the study mentioned above, quality of life was assessed using a 36-item medical outcomes study short-form health survey (SF-36). Physical Component Summary (PCS) and Mental Component Summary (MCS), derived from SF-36, were identified as predictors of mortality rates and hospitalizations in dialysis patients [6].

It is proven that physical activity affects HD patients’ quality of life, but studies comparing patients with healthy controls and investigating predictors of quality of life are lacking.

The present cross-sectional study aimed to evaluate and compare the quality of life and habitual physical activity of HD patients and healthy controls. It also aimed to investigate clinical predictors of quality of life in both study groups. It was hypothesized that HD patients would have lower levels of habitual physical activity and lower quality of life compared to healthy controls. In addition, we hypothesized that quality of life was related to age, body composition, comorbidities, and habitual physical activity of the subjects.

## 2. Materials and Methods

This was a cross-sectional study investigating the relationship between habitual physical activity and quality of life in HD patients and healthy controls.

### 2.1. Subjects

Ninety-three CKD (chronic kidney disease) patients undergoing hemodialysis at the dialysis units throughout Slovenia and 140 healthy controls took part in the research. The inclusion criteria for HD patients were a HD treatment duration of more than three months and an age of over 18 years. The control patients were included if they did not have renal disease and were over 18 years old. All subjects signed a written consent form to participate in the study. The research protocol was approved by the National Medical Ethics Committee (Ministry of Health, Republic of Slovenia, number of the approval document 125/05/14).

The demographic and clinical characteristics of the subjects are presented in Table 1. Unless otherwise stated, the data are presented as mean ± standard deviation (SD).

### 2.2. Procedure

Firstly, the questionnaire with information on demography, treatment, and comorbidities was given to all subjects. Secondly, we took their anthropometric body measurements and carried out an analysis of their body composition. The anthropometric measurements were assessed with instruments from SiberHagner (Zurich, Switzerland) and a body composition analysis with Fresenius Body Composition Monitor (Fresenius Medical Care, Bad Homburg, Germany). Thirdly, the participants’ quality of life and physical function were assessed using a 36-item medical outcomes study short-form health survey version 2 (SF-36) and Human Activity Profile (HAP) questionnaires. The questionnaires were completed with the help of an interviewer.

### 2.3. HAP Questionnaire

The HAP questionnaire consists of 94 items representing different activities, ranging from very easy (1st—getting up from the bed and from the chair without help) to very strenuous activities (94th—running five kilometers in 30 min or less) [20]. Participants were asked to indicate whether they still performed the activity, whether they had stopped doing the activity, or whether they never performed the activity. From the answers, we calculated the maximum activity score (MAS) and the adjusted activity score (AAS). MAS represents the activity number with the highest metabolic equivalent of the task (MET) level. This score reflects the current maximum activity level of the respondent. In contrast, the ASS score reflects the respondent’s average daily activity level. ASS is calculated by subtracting from the MAS score the number of less strenuous activities (below the MAS score) that the respondent no longer performs. For example, if the respondent states that the most strenuous activity he/she still performs is “climbing 50 stairs”, which is number 74 in the HAP questionnaire, the MAS score would be 74. If he/she no longer performs five activities that are below the number 74 (climbing 50 stairs), the ASS score would be 69. The ASS score gives us a more accurate estimate of the range of activities performed and the presence of impairments.

### 2.4. SF-36 Questionnaire

The Medical Outcomes Study Short-Form 36-Item General Health Survey (SF-36) is a frequently used instrument for assessing individual well-being and general functional health [21,22,23,24]. The questionnaire can be used in different population groups and in clinical settings. It consists of eight subscales: physical functioning, general health, role functioning—physical, mental health, bodily pain, vitality, role functioning—emotional and social functioning. Scores on the scales range from 0 to 100, with a higher score indicating better functioning. These scores are then converted to z-scores. In our study, we examined the Physical Component Summary (PCS) and the Mental Component Summary (MCS). The PCS and MCS are calculated by compressing these eight subscales into these two components using factor analysis. The mean value of each component is 50 with SD of 10 in the general population of the United States [23].

### 2.5. Statistical Analysis

Data were analyzed using SPSS v. 20.0 (IBM Corporation, Armonk, NY, USA), and descriptive statistics (mean ± SD) were used to compute demographic, clinical characteristics, anthropometric and body analysis measurements, and the results of HAP and SF-36 questionnaires. A Kolmogorov–Smirnov test failed to reject normality, and homoscedasticity was confirmed by the visual inspection of a residual scatter plot. An independent T-test estimated the mean differences (95% confidence interval) between HD patients and healthy people in the SF-36 components (PCS and MCS) and the HAP AAS. If a Levene’s test yielded unequal variances, we used a T-test that is robust for unequal variances (with a Satterthwaite approximation for the degrees of freedom).

Prior to testing the study hypotheses, we created a correlation matrix of pairwise Pearson’s correlation coefficient (*r*) to assess bivariate linear relationships between outcomes (PCS and MCS) and predictors (gender, age, LTI, FTI, Davies comorbidity score, and AAS), and multicollinearity (*r* ≥ 0.8) [25]. Pearson’s r values of ±0.10, ±0.30, and ±0.50 demarcated the weak, moderate, and strong relationships [26].

We attempted to model both SF-36 components, the PCS and MCS, but the MCS was not additionally analyzed since it did not significantly correlate with any of the observed variables. A hierarchical multiple linear regression including three sequential linear models tested whether the addition of LTI, FTI, and Davies comorbidity score improved the PCS prediction over and above age and gender alone. While controlling for those factors, we examined the HAP AAS influence on the PCS. Hence, we regressed the PCS on gender and age in the first model (Model 1), whereas the LTI, FTI, and Davies comorbidity score were added in the second model (Model 2). The full model (Model 3) additionally included the AAS. The same procedures were conducted for HD patients (Model_HD_) and healthy subjects (Model_H_). We reported unstandardized coefficients with 95% confidence intervals (B (95% CI)) from the linear models to depict relationships between predictors and each outcome, and standardized coefficients (β) to illustrate the relative contribution of each predictor to the outcome. The coefficient of determination (R^2^) adjusted for the number of predictors in the model (adjusted R^2^) was used to measure the goodness-of-fit of each model. The change in R^2^ and F (change statistics) from Model 1 to Model 2, and from Model 2 to Model 3, and the Akaike information criterion (AIC) were employed to compare the models within the groups. The level of significance was set at *p* ≤ 0.05.

## 3. Results

First, we examined the results from selected questionnaires separately for HD patients and for healthy controls. Table 2 shows mean ± SD results for each SF-36 component and for MAS and AAS. Healthy controls had a significantly higher mean score for the PCS and HAP AAS than HD patients.

For HD patients, the Pearson’s r revealed a strong significant positive correlation between PCS and HAP AAS (r_(91)_ = 0.605). A weak significant positive relationship was seen between PCS and LTI (r_(91)_ = 0.219) and inverse relationship between PCS and FTI (r_(91)_ = -0.211). We also observed that the PCS and HAP AAS tended to significantly decline with age. The negative correlation between age and HAP AAS (r_(91)_ = -0.518) was stronger than the correlation between age and PCS (r_(91)_ = −0.229).

The Pearson’s r established that healthy subjects also had a high degree of positive correlation between PCS and HAP AAS (r_(138)_ = 0.633). However, weak-to-moderate, significant negative correlations were perceived between PCS and FTI (r_(138)_ = −0.398), and PCS and Davies comorbidity score (r_(138)_ = −0.243). We also found a weak significant positive correlation between PCS and LTI (r_(138)_ = 0.292). For healthy controls, the PCS and HAP AAS also significantly decreased with age while observing a moderate-to-strong strength of the correlations with age (PCS: r_(138)_ = −0.477; HAP AAS: r_(138)_ = −0.653).

The MCS, however, did not significantly correlate with any of the observed variables in HD patients and healthy controls; therefore, it was not further analyzed. The correlation matrix showed no multicollinearity between used variables for HD patients and healthy controls. Results from the bivariate analysis are presented in Table 3.

### 3.1. Predictors of the Physical Component of Quality of Life (SF-36)

Three sequential linear models were used to model PCS for HD patients and healthy subjects. We regressed the PCS on gender and age in the first model (Model 1), and the LTI, FTI, and Davies comorbidity score were added in the second model (Model 2). The full model (Model 3) additionally included the AAS.

### 3.2. HD Patients

A hierarchical multiple linear regression yielded that the full model to predict the PCS was significant (Model_HD_ 3). However, the addition of LTI, FTI, and Davies comorbidity score (Model_HD_ 2) to the prediction of the PCS did not lead to a significant change in R^2^ (change statistics: R^2^ = 0.020, F_(3, 87)_ = 0.631, *p* = 0.597). The inclusion of the HAP AAS to predict the average PCS fitted the model (Model_HD_ 3) significantly better (change statistics: R^2^ = 0.320, F_(1, 86)_ = 45.995, *p* < 0.001). Hence, the HAP AAS accounted for 32% of the variation in the PCS after controlling for gender, age, LTI, FTI, and Davies comorbidity score effects. The predicted average increase in PCS scores was 0.431 for one extra activity adjusted score according to HAP, with all values of all other predictors being held constant. According to the AIC weight of the model, the best-fit model was Model_HD_ 3. Table 4 presents full details on each regression model.

### 3.3. Healthy Controls

A hierarchical multiple linear regression revealed that the full model to predict the PCS was significant (Model_H_ 3) in the subsample of healthy subjects. The addition of LTI, FTI, and Davies comorbidity score (Model_H_ 2) significantly improved the prediction of the PCS over and above age and gender alone (change statistics: R^2^ = 0.049, F_(3, 134)_ = 3.058, *p* = 0.031). However, the corresponding individual parameter coefficients were non-significant. Finally, the introduction of the HAP AAS to the PCS model predicted the data significantly better (change statistics: R^2^ = 0.149, F_(1, 133)_ = 34.679, *p* < 0.001). With all values of all other predictors being held constant, the estimated mean increase in the PCS scores was 0.359 for one extra activity adjusted score recorded by the HAP. Model_H_ 3 was found to be the best regression model, according to AIC. See Table 5 for full details on each regression model.

## 4. Discussion

A recent prospective cohort study conducted in 17 countries suggests that an increase in physical activity is associated with a reduction in the risk of mortality [27]. In addition, physical activity was found to be an independent predictor of mortality in HD patients [28]. Therefore, an increase in this population’s physical activity is very important for the quality of life and well-being [29]. However, many studies showed that HD patients are less physically active than the healthy sedentary population [1,30,31]. The main result of this study is that a level of habitual physical activity measured with the HAP questionnaire is correlated with quality of life in HD patients. Moreover, HD patients showed lower levels of habitual physical activity and lower quality of life compared to healthy controls. Additionally, habitual physical activity showed to be a better predictor of quality of life for HD patients than for healthy controls, which confirms our hypothesis.

Evaluating physical activity in HD patients has many difficulties [32]. Currently, there is no clear method for determining physical activity in HD patients. In terms of objective physical activity measurements, the accelerometer may underestimate the patient’s activity during low-intensity walking [33]. On the contrary, questionnaires are practical and inexpensive tools that showed good validity and reliability for assessing HD patients’ physical activity. Wong et al. [34] showed that none of the HD patients had high physical activity levels using the global physical activity questionnaire. Similar results were obtained using a three-dimensional accelerometer, which showed that HD patients were less active than healthy sedentary controls [1]. Interestingly, another study, also using a three-dimensional accelerometer, showed that HD patients were even less active than in previous studies [30].

Our study evaluated physical activity in HD patients using a HAP questionnaire that measures habitual physical activity. HD patients in the current study showed lowered scores for both AAS (HD patients: 73.24 ± 14.04 vs. healthy controls: 85.21 ± 9.00) and MAS (HD patients: 81.01 ± 8.92 vs. healthy controls: 87.54 ± 6.87) compared to healthy controls. Recently, Zhang et al. [28], using the HAP questionnaire, showed that lower MAS and AAS values were associated with a higher risk of death. Having in mind that the quality of life is reduced in HD patients compared to the general population, it was important to establish the relationship between habitual physical activity and quality of life. In the current study, habitual physical activity, measured by HAP, proved to be a significant predictor of quality of life (PCS of the SF-36). Specifically, it can be concluded that AAS can better predict the PCS of quality of life in HD patients than in healthy controls. Similar findings were found in Australians with CKD, where the activity level (HAP) supported the lower physical values of SF-36 [35]. The authors found that CKD patients showed significantly less ability to perform activities of daily living and were less able to engage in entertainment, social, or independent exercise compared to healthy people. This was confirmed by a systematic review that stated that self-reported physical activity is strongly associated with better health-related quality of life (HRQOL) and lower mortality in patients on HD [36]. Our results, along with other studies that showed the importance of physical activity for quality of life in HD patients, have significant value in designing the multicomponent lifestyle interventions involving physical exercise. This was confirmed by a novel study that showed that multi-component lifestyle intervention improved a variety of health outcomes in CKD patients [37]. Moreover, Mustata et al. [38] showed that long-term exercise training improved HRQOL in patients on predialysis, while Ota et al. [39] reported that a program including stretching and isotonic muscle conditioning improved components of daily living in elderly hemodialysis patients.

However, several limitations of the present study should be mentioned. Firstly, we used a questionnaire to determine physical activity, which is not considered as the gold standard to determine physical activity. However, questionnaires, as a multidimensional instrument, could assess different components of physical activity in HD patients. Furthermore, we did not use daily monitoring of physical activity to identify the possible influence of dialysis days on low physical activity. Another limitation of the study may be that the included HD patients involved twice as many males as females (62 males/31 females), but there were fewer males than females among the healthy controls (59 males/81 females). However, due to the fact that regression analysis showed no significant influence of gender on our outcomes, it may be assumed that the above-mentioned issue did not impact our conclusions. Finally, our results could not be generalized to non-dialysis and peritoneal dialysis CKD patients. However, the fact that we included several dialysis centers in Slovenia with a considerably large sample size represents the strength of our study.

## 5. Conclusions

In summary, the current results show that habitual physical activity is correlated with quality of life in HD patients. Moreover, this study confirms and extends previous findings that habitual physical activity and quality of life in HD patients are lower compared to healthy controls. It was also found that habitual physical activity is a better predictor of quality of life for HD patients than for healthy controls among the predictors analyzed. Therefore, interventions to increase physical activity in HD patients have the potential to be of great benefit, especially for their quality of life and well-being. Routine physical activity programs are therefore highly justified, and the nephrology community should play a leading role in this effort.

## Figures and Tables

**Table 1 ijerph-18-01978-t001:** Sample characteristics.

	HD Patients (*n* = 93)	Healthy Controls (*n* = 140)
Age (years)	55.04 ± 15.95	51.6 ± 16.1
Male/female (n)	62/31	59/81
Height (cm)	168.22 ± 9.04	170.1 ± 9.7
Weight (kg)	74.17 ± 15.71	75.5 ± 16.3
Weekly dialysis duration (h)	13.88 ± 2.55	n/a
LTI (kg/m^2^)	13.51 ± 2.73	13.73 ± 2.63
FTI (kg/m^2^)	11.99 ± 4.72	12.03 ± 5.12
Davies comorbidity score	0 (50.5%); 1 (49.5%)	0 (90.7%); 1 (9.3%)

Abbreviations: HD, hemodialysis; n, number of patients; LTI, lean tissue index; FTI, fat tissue index; n/a, not applicable. Davies comorbidity score (0 = no comorbidities; 1 = one or more comorbidities).

**Table 2 ijerph-18-01978-t002:** Results of the questionnaires.

	HD Patients (*n* = 93)	Healthy Controls (*n* = 140)	Mean Difference (95% CI)
SF36 PCS	48.1 ± 7.9	54.7 ± 6.1	6.6 (4.7, 8.5) ***
SF36 MCS	52.7 ± 8.1	54.0 ± 5.9	1.4 (0.5, 3.3)
HAP AAS	73.2 ± 14.0	85.2 ± 9.0	6.5 (4.5, 8.6) ***
HAP MAS	81.0 ± 8.9	87.5 ± 6.9	11.9 (9.0, 14.9) ***

Abbreviations: SF-36, the 36-item short-form health survey questionnaire; PCS, physical component summary; MCS, mental component summary; HAP AAS, human activity profile adjusted activity score; HAP MAS, human activity profile maximal activity score; n, number of subjects; *** significant at *p* < 0.001.

**Table 3 ijerph-18-01978-t003:** The Pearson’s r for observed variables.

	SF36 PCS	SF36 MCS	Gender	Age	HAP AAS	LTI	FTI	Davies Comorbidity Score
SF36 PCS	-	*−0.11*	*−0.13*	*−0.48 ***	*0.63 ***	*0.29 ***	*−0.40 ***	*−0.24 ***
SF36 MCS	0.03	-	*−0.04*	*0.13*	*0.00*	*0.06*	*0.07*	*−0.10*
gender	−0.08	0.06	-	*0.18 **	*−0.16 **	*−0.69 ***	*−0.69 ***	*−0.17 **
age	−0.23 *	0.05	−0.09	*-*	*−0.65 ***	*−0.38 ***	*−0.38 ***	*0.25 ***
HAP AAS	0.61 **	0.04	−0.22 *	−0.52 **	-	*0.32 ***	*−0.49 ***	*−0.21 ***
LTI	0.22 *	−0.04	−0.48 **	−0.38 **	0.58 **	-	*−0.43 ***	*−0.04*
FTI	−0.21 *	0.12	0.02	0.39 **	−0.42 **	−0.51 **	-	*0.17 **
Davies comorbidity score	−0.11	−0.06	−0.15	0.35 **	−0.222 *	−0.12	0.16	-

Note: Upper triangle in italics is the group of healthy controls (*n* = 140). The lower triangle is HD patients (*n* = 93). In these analyses, gender was coded by 1 for males and 2 for females. Abbreviations: SF-36, the 36-item short-form health survey questionnaire; PCS, physical component summary; MCS, mental component summary; HAP AAS, human activity profile adjusted activity score; LTI lean tissue index; FTI fat tissue index; * significant at *p* < 0.05; ** significant at *p* < 0.01.

**Table 4 ijerph-18-01978-t004:** Hierarchical multiple linear regression models of PCS (*n* = 93).

	PCS					
	Model_HD_ 1		Model_HD_ 2		Model_HD_ 3	
Predictors	B (95% CI)	β	B (95% CI)	β	B (95% CI)	Β
gender	−1.646 (−5.047, 1.755)	−0.099	−1.041 (−5.306, 3.224)	−0.062	−0.054 (−3.529, 3.421)	−0.003
age	−0.118 ^*^ (−0.219, 0.017)	−0.238	−0.075 (−0.197, 0.046)	−0.152	0.050 (−0.055, 0.156)	0.101
LTI			0.199 (−0.673, 1.071)	0.069	−0.621 (−1.368, 0.127)	−0.214
FTI			−0.183 (−0.606, 0.240)	−0.109	−0.073 (−0.418, 0.272)	−0.044
Davies comorbidity score			−0.654 (−4.120, 2.813)	−0.041	0.072 (−2.751, 2.895)	0.005
HAP AAS					0.431 (0.304, 0.557) ***	0.764
R^2^	0.062		0.082		0.402	
adjusted R^2^	0.041		0.029		0.360	
F	2.983		1.557		9.634 ^***^	
AIC	383.850		387.848		350.006	

Abbreviations: Model_HD_, model with hemodialysis patients; HAP AAS, human activity profile adjusted activity score; LTI, lean tissue index; FTI, fat tissue index; *B* (95% CI), unstandardized coefficient with 95% confidence intervals; β, standardized coefficient; R^2^, coefficient of determination; F, F statistic; AIC, Akaike information criterion; n, number of subjects; * significant at *p* < 0.05; *** significant at *p* < 0.001; Model_HD_ 1: α = 56.817; Model_HD_ 2: α = 53.490; Model_HD_ 3: α = 23.103.

**Table 5 ijerph-18-01978-t005:** Hierarchical multiple linear regression models of PCS (*n* = 140).

	PCS					
	Model_H_ 1		Model_H_ 2		Model_H_ 3	
Predictors	B (95% CI)	β	B (95% CI)	β	B (95% CI)	β
gender	−0.559 (−2.424, 1.306)	−0.045	0.309 (−2.285, 2.904)	0.025	0.290 (−2.029, 2.610)	0.023
age	−0.178 *** (−0.235, −0.121)	−0.469	−0.122 *** (−0.189, −0.054)	−0.321	−0.017 (−0.087, 0.053)	−0.044
LTI			0.255 (−0.264, 0.775)	0.110	0.210 (−0.255, 0.674)	0.090
FTI			−0.201 (−0.415, 0.013)	−0.168	−0.082 (−0.277, 0.114)	−0.068
Davies comorbidity score			−2.638 (−5.916, 0.640)	−0.125	−2.105 (−5.041, 0.831)	−0.100
HAP AAS					0.359 *** (0.239, 0.480)	0.525
R^2^	0.230		0.279		0.428	
adjusted R^2^	0.218		0.252		0.402	
F	20.411 ***		10.367 ***		16.590 ***	
AIC	476.066		472.796		442.358	

1 = males; 2 = females; Abbreviations: Model_H_, model with healthy controls; HAP AAS, human activity profile adjusted activity score; LTI, lean tissue index; FTI, fat tissue index; *B* (95% CI), unstandardized coefficient with 95% confidence intervals; β, standardized coefficient; R^2^, coefficient of determination; F, F statistic; AIC, Akaike information criterion; N number of subjects; *** significant at *p* < 0.001; Model_H_ 1: α = 64.728; Model_H_ 2: α = 59.629; Model_H_ 3: α = 22.783.

## Data Availability

Data generated and analyzed during this study are included in this article. Additional data are available from the corresponding author on request.

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
