# Peer review of "Physical Activity and Quality of Life in Hemodialysis Patients and Healthy Controls: A Cross-Sectional Study"

_ijerph, 2021, doi:10.3390/ijerph18041978_

Round 1

Reviewer 1 Report

The manuscript is well organized and well written. I only have minor suggestions.

  • The third paragraph of Discussion would need some improvement. The authors re-cited the numbers of their results too much in this section. It is redundant. Please only summarize your results briefly and what could be interpreted from our results in plain language. 
  • Authors should also review the literature more and compare their study with others that used HAP questionnaire (or at least other questionnaire) to this paragraph. Is your study similar or different from what previously reported? This would help highlight the significance of your study.
  • The second paragraph lacks key delivering message. I am not sure what are you trying to infer to. Please provide one or two sentence summarizing the point of this paragraph at the beginning or the end of the paragraph.
  • Under Limitations. Please elaborate the first two limitations further and provide us your rationalization. How a questionnaire could impact the validity of the study. Which type of bias is your study subjected to? What could be the impact of not using daily monitoring of physical activity.

Reviewer 2 Report

This work presents a cross-sectional study for evaluating the quality of life and habitual physical activity of Hemodialysis patients and healthy controls. It also investigates clinical predictors of quality of life in both study groups. The study results showed that physical activity in everyday life, as measured by Human Activity Profile questionnaire, is associated to a higher degree with the quality of life of Hemodialysis patients than in healthy subjects.

The work is relevant for the field, and the contribution is important. Nevertheless, some issues should be clarified before the manuscript could be considered for publication.

Comments:

1) The Hemodialysis patients (62 male/31 female) present double male compared to female subjects, but healthy controls (59 male/81 female) present fewer male than female.

The demographic and clinical characteristics of the subjects could generate a bias on the results.

2) Dialysis treatment could affect physical activity possibilities, producing a restriction.

3) Quality of life assessment could be influenced by the health state of the patient, generating a bias on the results.

4) The Conclusions section should provide quantitative results.

Round 2

Reviewer 2 Report

All the recommendations were addressed by the authors, and the questions were clarified, I have not further comments, and I believe the manuscript could be considered for publication.